# Emerging Immunotherapy Approaches for Advanced Clear Cell Renal Cell Carcinoma

**DOI:** 10.3390/cells13010034

**Published:** 2023-12-22

**Authors:** Lingbin Meng, Katharine A. Collier, Peng Wang, Zihai Li, Paul Monk, Amir Mortazavi, Zhiwei Hu, Daniel Spakowicz, Linghua Zheng, Yuanquan Yang

**Affiliations:** 1Division of Medical Oncology, Department of Internal Medicine, The Ohio State University Comprehensive Cancer Center, Columbus, OH 43210, USA; katharine.collier@osumc.edu (K.A.C.); peng.wang@osumc.edu (P.W.); zihai.li@osumc.edu (Z.L.); paul.monk@osumc.edu (P.M.); amir.mortazavi@osumc.edu (A.M.); daniel.spakowicz@osumc.edu (D.S.); linghua.zheng@osumc.edu (L.Z.); 2Pelotonia Institute for Immuno-Oncology, The Ohio State University Comprehensive Cancer Center, Columbus, OH 43210, USA; 3Division of Surgical Oncology, Department of Surgery, The Ohio State University Comprehensive Cancer Center, Columbus, OH 43210, USA; zhiwei.hu@osumc.edu

**Keywords:** clear cell renal cell carcinoma, immune checkpoint inhibitors, immunotherapeutic combinations, bispecific antibodies, CAR T cells, vaccines, cytokines, oncolytic virus

## Abstract

The most common subtype of renal cell carcinoma is clear cell renal cell carcinoma (ccRCC). While localized ccRCC can be cured with surgery, metastatic disease has a poor prognosis. Recently, immunotherapy has emerged as a promising approach for advanced ccRCC. This review provides a comprehensive overview of the evolving immunotherapeutic landscape for metastatic ccRCC. Immune checkpoint inhibitors (ICIs) like PD-1/PD-L1 and CTLA-4 inhibitors have demonstrated clinical efficacy as monotherapies and in combination regimens. Combination immunotherapies pairing ICIs with antiangiogenic agents, other immunomodulators, or novel therapeutic platforms such as bispecific antibodies and chimeric antigen receptor (CAR) T-cell therapy are areas of active research. Beyond the checkpoint blockade, additional modalities including therapeutic vaccines, cytokines, and oncolytic viruses are also being explored for ccRCC. This review discusses the mechanisms, major clinical trials, challenges, and future directions for these emerging immunotherapies. While current strategies have shown promise in improving patient outcomes, continued research is critical for expanding and optimizing immunotherapy approaches for advanced ccRCC. Realizing the full potential of immunotherapy will require elucidating mechanisms of response and resistance, developing predictive biomarkers, and rationally designing combination therapeutic regimens tailored to individual patients. Advances in immunotherapy carry immense promise for transforming the management of metastatic ccRCC.

## 1. Introduction

Renal cell carcinoma (RCC) constitutes a considerable burden on public health, with an estimated 81,800 new diagnoses and 14,890 mortality cases predicted for 2023 in the United States alone [1]. Over the recent years, the incidence of RCC has exhibited a consistent upward trend [2,3]. Among the various subtypes, clear cell renal cell carcinoma (ccRCC) predominates, accounting for approximately 70–80% of RCC cases [4,5], with the majority originating from the proximal convoluted tubule [6]. Conversely, the non-clear cell renal cell carcinomas (nccRCCs), encompassing entities such as papillary, chromophobe, translocation, and medullary RCC, as well as collecting duct carcinoma, comprise 20–30% of RCC and harbor distinct histopathological and molecular characteristics [7,8]. An early critical event in ccRCC pathogenesis is the mutation or inactivation of the von Hippel-Lindau (VHL) tumor suppressor gene [9], resulting in the aberrant accumulation of hypoxia-inducible factors (HIFs) and the activation of pro-angiogenic signaling cascades, notably the vascular endothelial growth factor (VEGF) pathway [9,10]. The standard treatment for localized ccRCC typically includes surgical removal, either radical or partial nephrectomy, with curative intent [11]. However, 20–30% of patients experience metastatic recurrence post-surgery [12]. In the metastatic setting, the main systemic treatments revolve around two core strategies: targeted therapy and immunotherapy [13,14].

Targeted regimens focus on critical signaling molecules and cascades driving RCC proliferation [13], such as the VEGF pathway which promotes tumor angiogenesis [15,16] and the mTOR pathway which governs pivotal cellular functions [17,18]. Though these therapies confer survival benefits, acquired resistance and adverse effects like fatigue, hypertension, and dermatologic toxicities can emerge [19]. Over the past decade, checkpoint inhibitor immunotherapies, targeting key T-cell modulators like PD-1, PD-L1, and CTLA-4, have demonstrated profound efficacy in ccRCC, both as monotherapies and combination regimens [20,21]. However, sizeable proportions of patients exhibit primary or secondary resistance during therapy [22], propelling interest in enhancing immunotherapy via combinatorial approaches or innovative modalities like adoptive cell transfer and cancer vaccines [23,24].

This review aims to provide a comprehensive overview of immunotherapy’s role in managing metastatic ccRCC. As illustrated in Figure 1, key areas of focus include checkpoint inhibitors, combination immunotherapies, bispecific antibodies, chimeric antigen receptor (CAR) T-cell therapy, vaccines, and oncolytic viruses. Our primary objective is to critically evaluate the evolving immunotherapeutic landscape for advanced RCC, assessing the potential of emerging therapies to address current clinical challenges and improve patient outcomes. This detailed analysis of novel treatments seeks to elucidate the current status of immunotherapies and anticipate their future trajectory in RCC therapy.

## 2. Immune Checkpoint Inhibitors

### 2.1. Background on PD-1/PD-L1 and CTLA-4 Pathways

Checkpoint inhibitors constitute a class of immunotherapy designed to target and inhibit specific proteins that suppress anti-tumor immune responses [25]. Among these, PD-1 and PD-L1 inhibitors have gained prominence and regulatory approval for treating ccRCC [26,27]. Nivolumab and pembrolizumab exemplify PD-1 inhibitors [28], while atezolizumab, avelumab, and durvalumab represent PD-L1 inhibitors [29,30]. These agents function by disrupting the interaction between PD-1 on T cells and PD-L1 expressed on certain types of immune cells and malignant cells [29]. PD-1 binding to PD-L1 transmits an inhibitory signal to the T cell to refrain from cytolytic activity [31]. Due to the high PD-L1 expression on numerous cancer cells facilitating immune evasion, the monoclonal-antibody-mediated blockade of the PD-1/PD-L1 axis enhances anti-tumor immunity [31].

CTLA-4 inhibitors constitute another class of checkpoint inhibitors that amplify anti-tumor immunity via CTLA-4 inhibition on T cells, with ipilimumab being a well-characterized example [32,33]. Both PD-1 and CTLA-4 are expressed on activated T cells and serve to attenuate T-cell responses upon binding their ligands [34]. Based on clinical trial outcomes and a regulatory review, agents like nivolumab, pembrolizumab, and ipilimumab have received US Food and Drug Administration (FDA) approval for treating various malignancies [33,34].

### 2.2. Immune Checkpoint Inhibitor Monotherapy in Advanced ccRCC

First-line therapy: Table 1 summarizes some key clinical trials of immune checkpoint inhibitor monotherapy in ccRCC. In the first-line setting for advanced ccRCC, monotherapy with immune checkpoint inhibitors has demonstrated heterogeneous outcomes. Two phase II clinical trials, KEYNOTE-427 and HCRN GU16-260, examined pembrolizumab and nivolumab as single agents, respectively.

KEYNOTE-427 evaluated pembrolizumab monotherapy in patients with locally advanced or metastatic RCC and reported an objective response rate (ORR) of 36.4% in the ccRCC cohort [35]. This ORR consisted of 4 complete responses and 36 partial responses. The median duration of response (DOR) was reported as 18.9 months, and 64.1% of patients sustained a response for at least 12 months. However, the median progression-free survival (PFS) was relatively short at 7.1 months, alongside a median overall survival (OS) of 40.7 months [35,36]. Thus, while pembrolizumab monotherapy exhibits modest efficacy, its overall therapeutic utility appears to be limited.

On the other hand, HCRN GU16-260 evaluated nivolumab monotherapy followed by salvage nivolumab/ipilimumab in 123 patients with treatment-naïve advanced ccRCC [37]. Nivolumab elicited objective responses across International Metastatic RCC Database Consortium (IMDC) risk categories, including a 57.1% ORR in favorable-risk patients. The overall ORR was 34.1% with a median DOR of 27.6 months. PD-L1 tumor expression was correlated with efficacy endpoints. However, salvage therapy with nivolumab/ipilimumab following disease progression on nivolumab monotherapy demonstrated limited benefit, with an ORR of 11.4%. Treatment-related adverse events were consistent with the known safety profiles of nivolumab and combination immunotherapy. In conclusion, nivolumab monotherapy showed clinically meaningful antitumor activity across IMDC risk groups in untreated ccRCC, with favorable-risk patients deriving substantial benefits [37].

Both trials revealed limited single-agent activity in treatment-naïve ccRCC. Consequently, immune checkpoint inhibitor monotherapy is not recommended as first-line therapy for stage IV ccRCC, with combination regimens being preferred.

Second-line therapy: In the second-line setting for advanced ccRCC, the role of monotherapy was notably investigated in the phase III CheckMate 025 trial, which compared nivolumab against everolimus in patients previously treated with anti-angiogenic therapy [38]. The results demonstrated superior efficacy for nivolumab, with significant improvements in OS and ORR. A five-year analysis with a median follow-up of 72 months underscored the durable advantage of nivolumab over everolimus in OS (25.8 vs. 19.7 months) and five-year OS rates (26% vs. 18%) [39]. Nivolumab also conferred a higher ORR (23% vs. 4%) and PFS (HR = 0.84, 95% CI: 0.72–0.99, *p* = 0.0331), although improvements in median PFS were not observed [39]. These findings have contributed to the acceptance of nivolumab monotherapy as a viable option for patients with advanced RCC following prior targeted therapy.

Adjuvant therapy: In the adjuvant setting, the phase III KEYNOTE-564 trial assessed pembrolizumab versus placebo after surgical resection in the high-risk ccRCC [40]. Pembrolizumab significantly prolonged disease-free survival compared to the placebo, representing the first phase III evidence for improved disease-free survival with adjuvant immunotherapy in this population. Although these results indicate promising efficacy for pembrolizumab, demonstrating an OS benefit is essential to fully ascertain its impact [40].

Safety and toxicities: It is critical to note the potential immune-related adverse events associated with checkpoint inhibitors, commonly including endocrinopathies, colitis, hepatitis, pneumonitis, rash, and fatigue [41,42]. Other concerns include infusion reactions, musculoskeletal pain, renal complications, and neurological toxicities [43]. In some cases, severe or even fatal immune-mediated toxicities, especially with PD-1/PD-L1 inhibitors, have been observed [44]. These adverse events underscore the necessity for a well-structured monitoring and management protocol to ensure patient safety while maximizing therapeutic benefits.

### 2.3. Exploration of Novel Immune-Modulating Therapies for Renal Cell Carcinoma

The therapeutic landscape for RCC is continuously evolving, with substantial efforts focused on developing novel agents that target immune checkpoint proteins or signaling pathways. Key areas of investigation include immune checkpoints such as lymphocyte-activation gene 3 (LAG-3) [45,46], T-cell immunoglobulin and mucin-domain containing-3 (TIM-3) [47,48], and T-cell immunoreceptor with immunoglobulin and tyrosine-based inhibitory motif domain (TIGIT) [49,50], alongside notable pathways including C-C motif chemokine ligand 2/C-C chemokine receptor type 2 (CCL2/CCR2) [51,52], Interleukin-1 (IL-1) [52], and Angiopoietin-2 (Ang2) [53]. The overarching objective is to enhance antitumor immune responses to improve patient outcomes.

A pivotal addition to this realm is a phase II trial (NCT05805501), which investigates the efficacy and safety of the combination of RO7247669 (PD1-LAG3) with axitinib, and, in some arms, the addition of tiragolumab (anti-TIGIT), for patients with untreated, locally advanced unresectable or metastatic ccRCC. This study aims to enhance our understanding of the potential of immune checkpoint inhibitors in treating RCC, pushing forward the frontier of combination therapies to improve antitumor immune responses and patient outcomes.

Another noteworthy initiative is a phase I/II clinical trial evaluating the combination of MEDI9197, a TLR 7/8 agonist, with durvalumab, an anti-PD-L1 inhibitor [54]. This study includes patients with advanced solid tumors, including ccRCC, and primarily aims to assess the tolerability and potential effectiveness of this regimen in enhancing antitumor immunity [54]. In parallel, another phase I/II study is investigating the synergistic combination of NKTR-214, a CD122-biased agonist, with the anti-PD-1 agent nivolumab [55]. This trial also enrolls patients with advanced solid tumors such as RCC to determine the tolerability and potential benefits of this approach in bolstering antitumor immune responses.

Indeed, numerous clinical studies are underway, each evaluating novel agents targeting unique immune checkpoints or exploring related signaling pathways, with a focus on ccRCC. As these trials progress, they are collectively advancing a dynamic field holding promise for improved therapeutic strategies to provide a clinical benefit for patients with this challenging malignancy.

## 3. Combination Immunotherapies

### 3.1. Rationale for Combining Immunotherapies

Checkpoint inhibitors targeting PD-1/PD-L1 and CTLA-4 have significantly advanced treatment options for metastatic ccRCC [25]. However, many patients exhibit resistance initially or relapse following monotherapy [56,57]. Both preclinical and clinical evidence suggest the potential utility of combination strategies, either pairing checkpoint inhibitors or integrating agents such as VEGF inhibitors or immunomodulators [58,59]. These combinatorial regimens aim to exert a broader anticancer effect, with goals to improve response rates, extend response duration, and overcome resistance mechanisms in immunogenic tumors. The synergy observed in these combinations could markedly alter therapeutic approaches for metastatic ccRCC [58]. (See Table 2 for a summary of key studies.)

### 3.2. PD-1/PD-L1 Inhibitors + CTLA-4 Inhibitors

The efficacy of this combination strategy was solidified in the phase III CheckMate 214 trial [60]. This first-line study in untreated metastatic ccRCC patients compared nivolumab/ipilimumab to sunitinib and demonstrated notable improvements in OS, ORR, and PFS versus sunitinib, particularly among intermediate/poor-risk patients. The combination therapy yielded an ORR of 42% compared to 27% with sunitinib. Complete response rates were 11% and 2% for the combination and sunitinib arms, respectively. The combination also showed superior median PFS (11.6 vs. 8.4 months with sunitinib). These pivotal findings have underpinned guidelines endorsing nivolumab/ipilimumab as a first-line therapy for intermediate/poor-risk metastatic ccRCC [60].

An extended five-year follow-up from CheckMate 214 further reinforced the combinatorial efficacy [61]. OS was longer for nivolumab/ipilimumab versus sunitinib (47.0 vs. 26.6 months), with five-year survival rates of 43% and 31%, respectively. The combination therapy also maintained superior ORR (42% vs. 27%) and complete response rates (11% vs. 2%) over sunitinib. More patients on combination therapy achieved complete responses without subsequent progression (9.6% vs. 2.4%). While the median DOR was unreached for nivolumab/ipilimumab, it was 24.8 months with sunitinib [61], underscoring the potential for durable responses with combination immunotherapy.

The nivolumab/ipilimumab combination was also assessed as a second-line therapy in the phase I CheckMate 016 trial [62]. Patients with metastatic ccRCC received nivolumab/ipilimumab followed by nivolumab maintenance. Severe toxicity precluded assessment of the nivolumab 3 mg/kg + ipilimumab 3 mg/kg cohort. In the nivolumab 3 mg/kg + ipilimumab 1 mg/kg and nivolumab 1 mg/kg + ipilimumab 3 mg/kg cohorts (n = 47 each), severe adverse events occurred in 38.3% and 61.7% of patients, respectively, alongside a 40.4% ORR in both groups. After a 22.3-month follow-up, ongoing responses were 42.1% and 36.8%. Two-year survival rates were 67.3% and 69.6% for the respective cohorts [62], revealing significant antitumor activity for second-line nivolumab/ipilimumab in metastatic ccRCC.

### 3.3. Checkpoint Inhibitors + VEGF Inhibitors

A promising therapeutic strategy in oncology entails the combination of immune checkpoint inhibitors with VEGF inhibitors, such as cabozantinib and axitinib [63]. VEGF inhibitors target the VEGF signaling cascade, a crucial mechanism in tumor angiogenesis [64]. By suppressing VEGF, angiogenesis may be hindered, thereby potentially restricting tumor growth. The combination of VEGF inhibitors with a checkpoint blockade could enhance antitumor immune responses and mitigate tumor-induced immunosuppression.

#### 3.3.1. Avelumab + Axitinib

The JAVELIN Renal 101 phase III trial evaluated the efficacy of first-line avelumab plus axitinib compared to sunitinib in patients with advanced ccRCC [65]. This study demonstrated pronounced improvements in PFS and ORR with combination therapy compared to sunitinib monotherapy for untreated metastatic ccRCC. Among 886 randomized participants, the subgroup with PD-L1-positive tumors exhibited a superior median PFS of 13.8 months with avelumab-axitinib versus 7.0 months with sunitinib. This PFS benefit persisted in the overall population, with medians of 13.3 and 8.0 months for the combination and sunitinib arms, respectively. The combination therapy also yielded a higher ORR of 53% compared to 27% for sunitinib [65]. An extended follow-up analysis demonstrated sustained improvements in survival, response rates, and DOR with the combination of avelumab and axitinib compared to sunitinib. The median OS was not reached for the combination of avelumab and axitinib, compared to 37.8 months for sunitinib. The median PFS was 13.9 months for avelumab and axitinib, versus 8.5 months for sunitinib [66]. In summary, avelumab-axitinib demonstrated enhanced efficacy and acceptable safety compared to sunitinib, although we await more comprehensive data.

#### 3.3.2. Pembrolizumab + Axitinib

The KEYNOTE-426 phase III trial evaluated the efficacy of combined pembrolizumab plus axitinib compared to sunitinib monotherapy in patients with previously untreated advanced or metastatic ccRCC [67]. Across 861 randomized participants, the treatment arms consisted of pembrolizumab-axitinib combination therapy versus sunitinib alone, with primary endpoints of OS, PFS, and ORR. After a median follow-up of 42.8 months, results demonstrated superior efficacy for the combination regimen. The median OS was not reached with pembrolizumab-axitinib, showing marked improvement compared to 35.7 months for sunitinib. The combination yielded a higher ORR of 59.3% versus 35.7% for sunitinib, alongside complete response rates of 5.8% and 1.9%, respectively. Combination therapy also conferred a superior median PFS of 15.4 months compared to 11.1 months with sunitinib [67]. These findings underlie the FDA approval of pembrolizumab-axitinib as a first-line therapy for advanced ccRCC.

After 67.2 months of extended follow-up (five-year analysis), pembrolizumab-axitinib maintained improved outcomes versus sunitinib for advanced ccRCC [68]. At 60 months, OS rates were 41.9% for pembrolizumab-axitinib and 37.1% for sunitinib, while PFS rates were 18.3% and 7.3%, respectively. The median DOR was longer with pembrolizumab-axitinib. Accounting for more subsequent therapies in the sunitinib arm, the OS advantage of pembrolizumab-axitinib remained significant [68]. Collectively, these data from KEYNOTE-426 demonstrate enhanced efficacy for pembrolizumab-axitinib combination therapy in advanced RCC [69].

#### 3.3.3. Nivolumab + Cabozantinib 

The phase III CheckMate 9ER trial evaluated the nivolumab plus cabozantinib combination therapy compared to sunitinib monotherapy in untreated patients with advanced or metastatic ccRCC [70]. This international, open-label, randomized study of 651 patients set PFS as the primary endpoint, assessed by an independent blinded review. After a median follow-up of 18.1 months, the nivolumab-cabozantinib combination demonstrated a superior median PFS of 16.6 months versus 8.3 months with sunitinib. Additionally, the combination conferred a markedly higher ORR of 55.7% compared to 27.1% for sunitinib. At one year, OS rates were 85.7% for the combination versus 75.6% with sunitinib. These compelling data endorse nivolumab-cabozantinib as a notable first-line strategy integrating immunotherapy and targeted therapy for metastatic ccRCC [70].

Extended three-year follow-up data showed durable advantages for nivolumab-cabozantinib over sunitinib, including a median PFS of 16.6 versus 8.4 months and median OS of 49.5 versus 35.5 months [71]. The combination therapy also maintained a higher ORR (56% vs. 28%) and complete response rate (13% vs. 5%). While adverse events were slightly increased with nivolumab-cabozantinib, no new safety signals emerged. These sustained benefits further endorse nivolumab-cabozantinib as a first-line therapy for advanced ccRCC [71].

#### 3.3.4. Pembrolizumab + Lenvatinib 

The phase III CLEAR trial assessed the combination therapy of lenvatinib and pembrolizumab compared to sunitinib monotherapy in untreated patients with advanced or metastatic ccRCC [72]. Across 1,069 randomized participants, the treatment arms consisted of lenvatinib-pembrolizumab, lenvatinib-everolimus, or sunitinib, with PFS as the primary endpoint. The lenvatinib-pembrolizumab combination demonstrated a superior median PFS of 23.9 months versus 9.2 months for sunitinib, alongside a higher ORR (71.0% vs. 36.1%) and complete response rates (16.1% vs. 4.2%). This combination also showed a pronounced OS advantage over sunitinib. Meanwhile, lenvatinib-everolimus conferred an improved median PFS of 14.7 months compared to 9.2 months with sunitinib [72].

In the extended follow-up from the CLEAR trial, lenvatinib-pembrolizumab maintained a superior PFS of 23.3 months versus 9.2 months for sunitinib after a median follow-up of 27.8 months [73]. Lenvatinib-pembrolizumab also exhibited prolonged OS (median not reached) compared to 33.7 months for sunitinib. Hazard ratios confirmed the significant benefits of the combination over sunitinib [73]. These durable efficacy data underscore the potential of lenvatinib-pembrolizumab as a first-line therapy for advanced ccRCC.

Separately, the phase Ib/II KEYNOTE-146 trial evaluated lenvatinib-pembrolizumab as a second-line therapy [74]. After initial dose-finding across multiple tumor types, the study focused on ccCC, with 65% of patients having received prior ICI and/or tyrosine kinase inhibitor (TKI) therapy. Early results indicated positive effects on PFS and response rates, supporting further research on lenvatinib-pembrolizumab in this setting [75].

### 3.4. Ongoing Clinical Trials of Combination Immunotherapies 

Numerous clinical trials are currently in progress with the aim of advancing therapeutic approaches for ccRCC by harnessing synergistic combination immunotherapies [58]. These studies are exploring diverse regimens encompassing checkpoint inhibitors, VEGF inhibitors, and innovative agents. As summarized in Table 3, notable investigations include the assessment of a personalized vaccine (NCT05269381), the enzymatic inhibitor valemetostat (NCT04388852), the IL-8 inhibitor (NCT04572451), and aldesleukin (NCT03260504). There is also increasing interest in the monoclonal antibody SRF388 (NCT04374877) and the integration of radiation therapy (NCT05327686). Additional trials are evaluating neoadjuvant checkpoint inhibitor combinations, including NCT04393350 and NCT05319015.

Examining select examples illustrates the intricate details and objectives. The phase I trial NCT03260504 is evaluating aldesleukin plus pembrolizumab for metastatic ccRCC, based on the premise that aldesleukin can potentiate anti-cancer immune responses which pembrolizumab may enhance by blocking immune evasion [76]. The phase I/Ib study NCT04374877 is pioneering SRF388, a monoclonal antibody targeting IL-27, first as a monotherapy in advanced solid tumors, and then in combination with pembrolizumab in specific malignancies like ccRCC to assess potential synergistic benefits [77]. Another phase II trial, NCT05319015, is investigating neoadjuvant lenvatinib-pembrolizumab prior to surgical resection in ccRCC with inferior vena cava invasion, with the goal of optimizing pre-surgical anti-tumor effects.

Collectively, these clinical initiatives highlight promising progress in ccRCC therapeutic development and the vast potential of combination immunotherapy. Beyond advancing treatment modalities, these studies embody resilience and the commitment to transforming renal cancer care through pioneering research and cross-disciplinary collaboration. The breadth of strategies under exploration indicates a steadfast momentum to reshape the future landscape of ccRCC therapy.

## 4. Bispecific Antibodies Targeting T-Cell Costimulatory Receptors

### 4.1. Background on Bispecific Antibodies 

Bispecific antibodies, an emerging avenue in immunotherapy, possess the unique ability to bind two distinct antigens simultaneously [78,79]. Their dual-targeting potential shows particular promise for ccRCC, where these agents can be engineered to bridge interactions between cancerous and specific immune cells to amplify anti-tumor immunity [80].

### 4.2. Bispecific Antibodies in Development for ccRCC

A notable example in ccRCC is the chimeric bispecific G250/anti-CD3 monoclonal antibody [81]. The G250 monoclonal antibody displays selective binding activity towards an antigen present on most subtypes of RCC, while lacking affinity for normal tissues, aside from expression in the gastric mucosa and large bile ducts [82]. This bispecific antibody, with both G250 and anti-CD3 specificity, can connect G250 antigen-presenting RCC cells to T cells, aiding in the elimination of target cells. Historically, the utility of murine antibodies was limited due to the induction of anti-antibody immune responses, but chimeric antibodies can circumvent this issue by combining murine variable regions with human constant regions [81]. An alternative approach employed a biologically derived bispecific monoclonal antibody (bs-mAb) targeting the RCC-associated G250 antigen and indium-labeled diethylenetriaminepentaacetic acid (bs-mAb: G250xDTIn-1) [83]. The testing, conducted on RCC xenografts in mice, involved initial treatment with G250xDTIn-1, followed by the administration of intravenous 111In-labeled peptide. Although radiolabel uptake was notable in all tumors, it did not consistently correlate with G250 expression. Kinetic heterogeneity among the tumors suggested physiological factors like vascularity and permeability may be influential. These studies underscored that antigen expression alone does not fully predict pre-targeting efficacy.

The clinical evaluation of bispecific checkpoint inhibitors has also commenced, including a phase I/II trial of MEDI5752, a PD-1/CTLA-4-targeting monovalent bispecific antibody [84]. Early results (NCT03530397) indicated promising antitumor activity across advanced solid tumors. Doses below 1500 mg demonstrated better tolerability compared to higher doses, although RCC patients exhibited more discontinuations. Ongoing studies are evaluating the risk/benefit profile of MEDI5752, particularly at lower doses in RCC, based on its potential as a reduced toxicity alternative to CTLA-4 inhibitors [85]. 

Adding to this innovative landscape, the Xencor XmAb819-01 trial (NCT05433142) is examining the efficacy of XmAb819, a novel bispecific antibody, in combination with pembrolizumab for ccRCC [86]. XmAb819 was engineered as an XmAb 2+1 bispecific antibody, featuring two binding domains targeting ENPP3, a molecule often found on tumor cells, and one cytotoxic T-cell binding domain against CD3, a component of the T-cell receptor (TCR) complex. This design enables XmAb819 to simultaneously engage with both cancer cells and the immune system’s T-cells, potentially enhancing the immune response against the tumor. This trial is particularly significant as it includes patients with ccRCC, thus contributing to the body of research in this area. 

In summary, bispecific antibodies constitute an innovative therapeutic strategy in ccRCC, warranting active investigation to refine and optimize clinical outcomes as this area continues to evolve.

## 5. CAR T-Cell Therapy

### 5.1. Background on CAR T-Cell Approach

CAR T-cell therapy represents a groundbreaking immunotherapeutic approach that involves the custom engineering of a patient’s endogenous T cells to recognize and eliminate cancer cells [87]. Early successes have been most notable in select hematological malignancies [88]. However, ongoing investigation is evaluating the efficacy of this therapy in solid tumors, such as ccRCC [89], alongside advancements in creating solid tumor models to evaluate CAR T-cell activity [90]. A significant strategy in developing CAR T-cell therapy for RCC entails engineering specificity for antigens consistently expressed on RCC cells, with CD70 representing a primary target given that it is overexpressed across many ccRCC tumors [91].

### 5.2. CAR-T in Development for ccRCC 

As summarized in Table 4, multiple early-stage clinical trials are currently underway to assess the safety and therapeutic efficacy of CAR T-cell therapies for ccRCC [23,92]. A notable trial among these is TRAVERSE (NCT04696731), which is evaluating the allogeneic CD70-targeted CAR T-cell therapy, ALLO-316, in metastatic ccRCC patients refractory to both checkpoint inhibitors and TKIs [93,94]. Preliminary results have shown promising activity, with an 18% ORR and an 82% disease control rate (DCR) among 17 patients. Importantly, a subset with confirmed CD70+ disease achieved a 33% ORR and 100% DCR. To mitigate the risk of graft-versus-host disease (GVHD) associated with ALLO-316, modifications were made, including the disruption of the TCR alpha and a CD52 knockout, which allowed for ALLO-647 (anti-CD52 antibody) to deplete host T-cells and promote the persistence of allogeneic CAR T-cells [93]. Patients were administered escalating doses of ALLO-316, ranging from 40 to 120 million CAR T-cells (with a potential increase to 240 million), following lymphodepletion with fludarabine/cyclophosphamide, with or without the inclusion of ALLO-647. Overall, ALLO-316 exhibited a manageable safety profile, with 65% of patients experiencing cytokine release syndrome, but no instances of GVHD or immune effector cell-associated neurotoxicity syndrome (ICANS) were observed. The maximum tolerated dose remains under investigation.

Separately, an ongoing phase II trial (NCT03393936) is evaluating two RCC-targeting CAR T-cell products, CCT301-59 and CCT301-38, for stage IV metastatic ccRCC patients [95]. These CAR T cells are designed to recognize receptor tyrosine kinase-like orphan receptor 2 (ROR2) and AXL, respectively. ROR2 is known to regulate mitosis and migration in RCC [96], while AXL interacts with growth-arrest-specific protein 6 (Gas6) [97]. Prior to the infusion of CAR T cells, patients undergo lymphodepletion, and the CAR T cells are administered at escalating doses following a three-plus-three design.

Another phase I trial, COBALT-RCC (NCT04438083), is investigating CTX130, an experimental allogeneic anti-CD70 CAR T-cell therapy engineered using CRISPR/Cas9 gene editing to enhance specificity and safety, for the treatment of advanced ccRCC [98]. Initial findings presented at the 2022 Society for Immunotherapy of Cancer (SITC) Annual Meeting indicated that CTX130 was well-tolerated with preliminary evidence of antitumor activity. The trial incorporates a dose escalation phase (Part A) to determine the maximum tolerated dose, and a cohort expansion phase (Part B) to further evaluate the safety and efficacy. Patients underwent lymphodepletion with fludarabine and cyclophosphamide prior to CTX130 infusion on day +1. Among 13 evaluable patients, an ORR of 8% and a stable disease rate of 69% were observed, conferring a DCR of 77%. CTX130 exhibited a favorable safety profile across all dose levels tested, with 7 out of the 14 treated patients experiencing only grade 1/2 cytokine release syndrome. While 3 patients experienced severe infections, these were deemed unrelated to CTX130. Ongoing investigation aims to determine the optimal tolerated dose.

Additional trials are evaluating CAR T cells targeting tumor antigens like c-MET (NCT03638206) or the MUC1 C-terminus (NCT05239143) [99,100]. Collectively, these studies explore the potential of CAR T-cell therapy for advanced RCC, although challenges such as immunosuppressive tumor microenvironments, tumor lysis syndrome, anaphylaxis, neurotoxicity, cytokine release syndrome, and potential on-target, off-tumor toxicities persist [23]. Current research efforts are directed towards refining CAR T-cell therapies through strategies like dual antigen targeting or a combination with checkpoint inhibitors to enhance outcomes in ccRCC.

In summary, CAR T-cell therapy presents a promising transformative immunotherapeutic approach for ccRCC, building on innovative methodologies and early successes. Despite existing challenges, continued research and clinical trials are underway to determine the optimal utilization of this potent platform, aiming for improved patient outcomes. 

## 6. Other Immune-Based Therapies

Beyond checkpoint inhibitors, bispecific antibodies, and CAR T-cell therapies, additional immunotherapeutic approaches are being investigated for ccRCC treatment. Notable strategies under evaluation include therapeutic vaccines, cytokines, and oncolytic viruses.

### 6.1. Vaccines 

Vaccine immunotherapy is emerging as a potential therapeutic approach for RCC [101]. Table 5 summarizes some representative early-phase vaccine trials in RCC. A notable phase II study (NCT00031564) evaluated a B7-1 gene-modified autologous tumor cell vaccine combined with interleukin-2 (IL-2) in stage IV RCC [102]. Conducted by the H. Lee Moffitt Cancer Center and Research Institute with collaborators including the National Cancer Institute (NCI) and Chiron Corporation, this study aimed to assess the impact of combining vaccine therapy with IL-2 in patients with stage IV RCC. The trial had multiple objectives: determining tumor response rates, evaluating immunogenicity, tracking OS, and characterizing local and systemic toxicity across 49 enrolled participants. While this study has concluded, the results remain forthcoming.

Another trial, NCT00458536, is investigating a dendritic cell–renal cell carcinoma fusion vaccine plus granulocyte macrophage colony stimulating factor (GM-CSF) for the treatment of RCC. Conducted by the Beth Israel Deaconess Medical Center and sponsored by the NCI, this study aims to evaluate the safety profile and characterize the side effects of the vaccine when combined with GM-CSF, a biotherapeutic agent. Though currently active, the trial has closed enrollment.

While immunotherapy has transformed the treatment landscape for kidney cancer, challenges remain, particularly for stage IV disease. One potential strategy to enhance RCC vaccine efficacy is integration with established therapies like sunitinib or immune checkpoint inhibitors to promote robust immune activation while mitigating tumor-mediated immunosuppression [103,104].

In summary, despite promising preliminary data, numerous challenges persist in realizing the full potential of vaccine immunotherapy for RCC. Although no RCC-specific cancer vaccines have achieved regulatory approval to date, clinical trials remain imperative to inform future therapeutic directions for this approach. Ongoing efforts are focused on improving vaccine effectiveness through new target discovery and strategic combinations to seamlessly integrate with existing treatments.

### 6.2. Cytokines

Cytokine therapy, employing agents such as IL-2 and interferon-α (IFN-α), has emerged as a recognized therapeutic avenue for metastatic RCC [105,106,107]. A notable clinical trial, identified as NCT00554515, was established to evaluate the effectiveness of this treatment approach [108]. This trial aimed to discern if the response rate to high-dose IL-2 in metastatic RCC patients, who had “good” pathologic predictive features, exceeded that of an unselected historical patient group. Surprisingly, the results indicated that high-dose IL-2 induced durable remissions and prolonged survival across diverse risk categories, including both “good” and “poor-risk” patients.

Historically, before the advent of targeted therapies, advanced RCC treatment heavily relied on cytokine immunotherapy using either interferon or IL-2 [109]. Although this method occasionally led to long-term remission, it was not uniformly beneficial. A significant number of patients did not experience substantial improvement, and many encountered severe adverse reactions, a byproduct of the immunotherapy [110].

### 6.3. Adoptive Cell Transfer

Current research also includes numerous clinical trials investigating adoptive cell transfer (ACT) methodologies in ccRCC. This strategy involves the introduction of ex vivo activated immune cells such as lymphokine-activated killer (LAK) cells, cytokine-induced killer (CIK) cells, or tumor-infiltrating lymphocytes (TILs) [111]. Recent explorations into combining CIK cells with targeted therapies have yielded optimistic outcomes in RCC treatment.

An illustrative case reported by Yang et al. detailed a stage III clear cell RCC patient treated with autologous CIK cells and sorafenib following surgery [112]. Despite initial severe side effects from sorafenib, a reduced dose alongside CIK cell therapy led to a stable disease and improved quality of life. In addition, a retrospective study by Mai et al. assessed 34 patients with metastatic RCC, comparing the efficacy of sunitinib or sorafenib alone versus in combination with dendritic cell–CIK (DC-CIK) cells [113]. The combination therapy showed significantly better PFS and three-year OS, with fewer disease progressions and deaths. A phase I trial also examined the safety and efficacy of pembrolizumab, a PD-1 inhibitor, combined with DC-CIK cells in various solid tumors, including RCC [114]. This combination led to a median OS of 270 days and PFS of 162 days, with a 75% disease control rate in RCC patients. Adverse reactions were generally reversible and manageable.

Building upon this momentum, chimeric antigen receptor natural killer (CAR-NK) and chimeric antigen receptor natural killer T (CAR-NKT) cells are being recognized as formidable additions to the ACT spectrum [115,116,117]. Their inherent capabilities for immediate tumor recognition and destruction, coupled with a lower likelihood of adverse effects, mark them as promising contenders for future ccRCC therapies [118,119]. The exploration of these cell-based modalities could greatly influence the next generation of treatments for ccRCC, enhancing efficacy while minimizing toxicity.

Such innovative ACT approaches represent a promising avenue in the treatment of ccRCC, offering the potential for more effective and targeted immune responses against cancer cells. The integration of these novel therapies could potentially redefine the therapeutic landscape for patients with ccRCC.

### 6.4. Oncolytic Viruses

Oncolytic viruses constitute a groundbreaking immunotherapy modality designed to selectively target and destroy cancer cells, including ccRCC [120]. Both naturally occurring and bioengineered viruses are under investigation, with the latter involving the modification of native viral structures [121].

In preclinical studies, these agents have demonstrated the potential to precisely eradicate malignant cells while sparing healthy tissues [122,123]. For example, pioneering work at the Mayo Clinic’s Center for Individualized Medicine combines oncolytic viruses with CAR T-cell therapy, aiming to enhance precision against solid tumors [124].

Currently, multiple early-phase trials are underway, assessing the safety and efficacy of oncolytic viral therapy for ccRCC [125,126]. However, realizing the full potential of oncolytic virotherapy faces complex challenges, such as the immunosuppressive tumor microenvironment that can impair viral infection and cancer cell eradication. [120]. Additionally, concerns exist regarding inadvertent transgene expression in healthy tissues [127].

To address these hurdles, studies are exploring strategies to optimize oncolytic viral therapy for ccRCC, such as a combination with immune checkpoint inhibitors to boost therapeutic effects [128,129]. Additionally, the viral-mediated modulation of the tumor microenvironment represents a tactic to potentially augment T-cell functionality against solid tumors [128].

In summary, despite challenges, oncolytic viruses are a promising therapeutic avenue in ccRCC. Ongoing research and clinical trials play a critical role in determining the optimal applications of this innovative modality to benefit ccRCC patients. The continued momentum in this field underscores the significant potential of oncolytic viruses in ccRCC treatment.

## 7. Conclusions

The therapeutic landscape for ccRCC is undergoing a transformative shift, with numerous immunotherapies emerging as promising modalities. Notable approaches include combinations of immune checkpoint inhibitors with other agents, such as bispecific antibodies, chimeric antigen receptor T-cell therapy, cancer vaccines, oncolytic viruses, and cytokines. Despite early clinical promise, addressing the inherent challenges and limitations of these therapies remains imperative.

A palpable sense of optimism surrounds the potential of immunotherapy in ccRCC. Contemporary research is focused on rationally designed combinations, predictive biomarker development, and refining T-cell efficacy. A deep understanding of ccRCC immunobiology will inform customized therapeutic strategies tailored to the unique immunologic milieu of each tumor.

While progress is encouraging, sustained rigorous research remains critical. Identifying predictive biomarkers is paramount for guiding patient-specific treatment selection. The robust validation of emerging biomarkers in large cohorts is equally vital. Elucidating the mechanisms of innate and acquired resistance is also imperative, enabling strategic prevention to maximize treatment efficacy.

In summary, the rise of immunotherapy as a transformative force in ccRCC is undeniable. Realizing the full potential of these promising therapies demands continued collaborative efforts, translating insights from bench to bedside, and careful attention to ethical considerations. As we unravel the intricate interplay between ccRCC and the immune system, we edge closer to a new era of groundbreaking immunotherapies that instill hope in patients battling this challenging disease.

## Figures and Tables

**Figure 1 cells-13-00034-f001:**
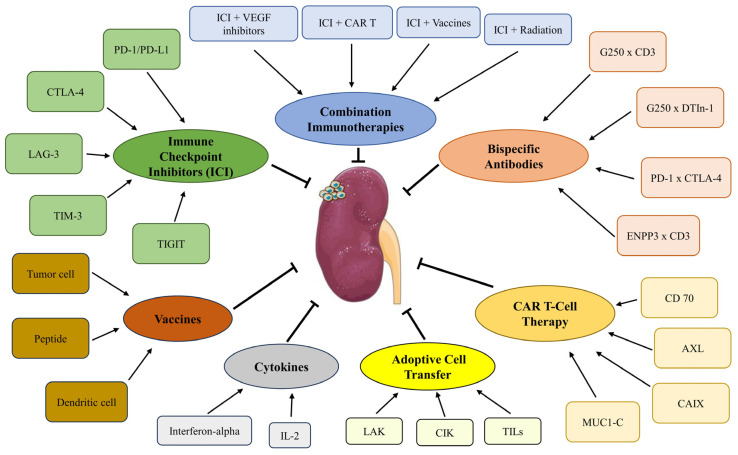
Overview of emerging immunotherapy approaches for ccRCC.

**Table 1 cells-13-00034-t001:** Key clinical trials of immune checkpoint inhibitor monotherapy in ccRCC.

NCT Number	Trial Name	Phase	Therapy Setting	Patients	Description	mOS (Months)	mPFS (Months)	ORR (%)
NCT02853344	KEYNOTE-427	2	1st line	110	pembrolizumab in locally advanced or metastatic ccRCC	40.7 (95% CI, 31.1–52.6)	7.1 (95% CI, 5.6–11.0)	36 (95% CI, 27–46)
NCT03117309	HCRN: GU16-260	2	1st line	123	nivolumab and salvage nivolumab + ipilimumab in advanced ccRCC	NR	8.3 (95% CI, 5.5–10.9)	34 (95% CI, 26–43)
NCT01668784	CheckMate 025	3	2nd line	410 vs. 411	nivolumab vs. everolimus in pretreated ccRCC	25.8 (95% CI, 22.2–29.8) vs. 19.7 (95% CI, 17.6–23.1); HR = 0.73 (95% CI, 0.62–0.85), *p* < 0.0001	4.2 (95% CI, 3.7–5.4) vs. 4.5 (95% CI, 3.7–5.5) HR = 0.84 (95% CI, 0.72–0.99), *p* = 0.0331	23 (95% CI, 19–27) vs. 4 (95% CI, 2–7)
NCT02420821	KEYNOTE-564	3	Adjuvant	496 vs. 498	pembrolizumab vs. placebo in ccRCC post nephrectomy	NR	NR	NR

Abbreviations: CI: confidence interval; HR: hazard ratio; mOS: median overall survival; mPFS: median progression-free survival; NCT: National Clinical Trial; NR: not reached; ORR: objective response rate.

**Table 2 cells-13-00034-t002:** Key clinical trials of immune checkpoint inhibitor combinations in ccRCC.

NCT Number	Trial Name	Phase	Therapy Setting	Patients	Description	mOS (Months)	mPFS (Months)	ORR (%)
NCT02231749	CheckMate 214	3	1st line	425 vs. 422	nivolumab + ipilimumab vs. sunitinib	47.0 (95% CI, 35.4–57.4) vs. 26.6 (95% CI, 22.1–33.5); HR = 0.68 (95% CI, 0.58–0.81), *p* < 0.0001	11.6 (95% CI, 8.7–15.5) vs. 8.4 (95% CI, 7.0–10.8) HR = 0.82 (99.1% CI, 0.64–1.05), *p* = 0.03	42 (95% CI, 37–47) vs. 27 (95% CI, 22–31)
NCT01472081	CheckMate 016	1	2nd line	47 vs. 47	3 mg/kg nivolumab + 1 mg/kg ipilimumab vs. 1 mg/kg nivolumab + 3 mg/kg ipilimumab	NR (95% CI, 26.7-NE) vs. 32.6 (95% CI, 26.0-NE)	7.7 (95% CI, 3.7–14.3) vs. 9.4 (95% CI, 5.6–18.6)	40 (95% CI, 26–56) vs. 40 (95% CI, 26–56)
NCT02684006	Javelin Renal 101	3	1st line	442 vs. 444	avelumab + axitinib vs. sunitinib	NE (95% CI, 30-NE) vs. NE (95% CI, 27.4-NE) HR = 0.80 (95% CI, 0.62–1.03), *p* = 0.0392	13.3 (95% CI, 11.1–15.3) vs. 8.0 (95% CI, 6.7–9.8) HR = 0.69 (95% CI, 0.57–0.83), *p* < 0.0001	53 (95% CI, 48–57) vs. 27 (95% CI, 23–32)
NCT02853331	KEYNOTE-426	3	1st line	432 vs. 429	pembrolizumab + axitinib vs. sunitinib	NR vs. 35.7 (95% CI, 33.3-NE) HR = 0.53 (95% CI, 0.38–0.74), *p* < 0.0001	15.4 (95% CI, 12.7–18.9) vs. 11.1 (95% CI, 9.1–12.5) HR = 0.71 (99.8% CI, 0.60–0.84), *p* < 0.0001	59 (95% CI, 55–64) vs. 36 (95% CI, 31–40)
NCT03141177	CheckMate 9ER	3	1st line	323 vs. 328	carbozantinib + nivolumab vs. sunitinib	NR vs. NR HR = 0.60 (98.9% CI, 0.40–0.89), *p* = 0.001	16.6 (95% CI, 12.5–24.9) vs. 8.3 (95% CI, 7.0–9.7) HR = 0.51 (95% CI, 0.41–0.64), *p* < 0.0001	56 (95% CI, 50–61) vs. 27 (95% CI, 22–32)
NCT02811861	Clear	3	1st line	355 vs. 357	lenvatinib + pembrolizumab vs. sunitinib	NR vs. NR HR = 0.66 (95% CI, 0.49–0.88), *p* = 0.005	23.9 (95% CI, 20.8–27.7) vs. 9.2 (95% CI, 6.0–11.0) HR = 0.39 (95% CI, 0.32–0.49), *p* < 0.001	71 (95% CI, 66–76) vs. 36 (95% CI, 48–59)
NCT02420821	Immotion 151	3	1st line	454 vs. 461	atezolizumab + bevacizumab vs. sunitinib	36.1 (95% CI, 31.5–42.3) vs. 35.3 (95% CI, 28.6–42.1) HR = 0.0.91 (95% CI, 0.76–1.08), *p* = 0.27	9.6 (95% CI, 8.3–11.5) vs. 8.3 (95% CI, 7.0–9.7) HR = 0.88 (95% CI, 0.74–1.04), *p* = 0.12	37 (95% CI, 32–41) vs. 33 (95% CI, 29–38)
NCT02501096	KEYNOTE-146	1b/2	2nd line	145	lenvatinib + pembrolizumab	32.2 (95% CI, 29.8–55.8)	14.1 (95% CI, 11.6–18.4)	63 (95% CI, 55–71)

Abbreviations: CI: confidence interval; HR: hazard ratio; mOS: median overall survival; mPFS: median progression-free survival; NCT: National Clinical Trial; NE: not estimable; NR: not reached; ORR: objective response rate.

**Table 3 cells-13-00034-t003:** Key ongoing clinical trials of combination immunotherapies in advanced or metastatic RCC.

NCT Number	Trial Name	Phase	Estimated Patients	Description	Sponsor
NCT05269381	PNeoVCA	1	36	pembrolizumab + personalized neoantigen peptide vaccine	Mayo Clinic
NCT04388852	NA	1	80	ipilimumab + valemetostat	M.D. Anderson Cancer Center
NCT04572451	NA	1	50	nivolumab + anti IL-8 + SBRT	University of Pittsburgh
NCT03260504	NA	1	15	pembrolizumab + aldesleukin	University of Washington
NCT04374877	KEYNOTE-C16	1	220	pembrolizumab + anti IL-27	Surface Oncology
NCT05327686	SAMURAI	2	240	nivolumab or pembrolizumab + axitinib + cabozantinib + SBRT	NRG Oncology
NCT04393350	NA	2	22	pembrolizumab + perioperative lenvatinib	Emory University
NCT05319015	NA	2	30	pembrolizumab + neoadjuvant lenvatinib	UTSW
NCT02811861	KEYNOTE-581	3	1069	levatinib + everolimus or pembrolizumab	Eisai Inc.
NCT05361720	OPTIC	2	54	ipilimumab + nivolumab + cabozantinib	Vanderbilt-Ingram Cancer Center
NCT03288532	RAMPART	3	1750	durvalumab + tremelimumab	University College, London
NCT05148546	NESCIO	2	69	neoadjuvant nivolumab+ ipilimumab + relatlimab	The Netherlands Cancer Institute
NCT04322955	Cyto-KIK	2	48	nivolumab + cabozantinib + cytoreductive nephrectomy	Columbia University
NCT05188118	NA	1	20	ipilimumab + nivolumab + cabozantinib	Icahn School of Medicine at Mount Sinai
NCT05363631	NA	1/2	55	pembrolizumab + axitinib + selenomethionine	University of Iowa
NCT04981509	NA	2	65	bevacizumab + erlotinib + atezolizumab	National Cancer Institute
NCT04090710	CYTOSHRINK	2	78	ipilimumab + nivolumab + SBRT	Ontario Clinical Oncology Group
NCT05411081	PAPMET2	2	200	atezolizumab + cabozantinib	National Cancer Institute

Abbreviations: IL: interleukin; NA: not available; NCT: National Clinical Trial; NRG: non-profit research organization; SBRT: stereotactic body radiotherapy; UTSW: University of Texas Southwestern Medical Center.

**Table 4 cells-13-00034-t004:** Representative early phase CAR-T trials in advanced or metastatic RCC.

NCT Number	Trial Name	Phase	Estimated Patients	Description	Sponsor
NCT04696731	TRAVERSE	1	120	CD-70 CAR-T (ALLO-316) in advanced or metastatic ccRCC	Allogene Therapeutics
NCT03393936	NA	1/2	66	AXL CAR-T (CCT301-38) or ROR2 CAR-T (CCT301-59) in recurrent or refractory stage IV RCC	Shanghai PerHum Therapeutics Co., Ltd.
NCT04438083	COBALT-RCC	1	107	CD70 CAR-T (CTX130) in relapsed, or refractory ccRCC	CRISPR Therapeutics AG
NCT03638206	NA	1/2	73	multi-target CAR-T/TCR-T in various malignancies, including c-MET CAR-T in ccRCC	Shenzhen BinDeBio Ltd.
NCT05239143	NA	1	100	MUC1-C CAR-T in advanced or metastatic solid tumors, including ccRCC	Poseida Therapeutics, Inc.
NCT05420519	PBC036	1	24	CD70 CAR-T in advanced or metastatic RCC	Chongqing Precision Biotech Co., Ltd.
NCT04969354	NA	1	20	CAIX CAR-T in advanced RCC	The Affiliated Hospital of Xuzhou Medical University
NCT05518253	PBC038	1	36	CD70 CAR-T in CD70-positive advanced or metastatic solid tumors, including ccRCC	Zhejiang University
NCT05468190	PBC041	1	48	CD70 CAR-T in CD70-positive advanced or metastatic solid tumors, including ccRCC	Chongqing Precision Biotech Co., Ltd.
NCT05420545	PBC037	1	36	CD70 CAR-T in CD70-positive advanced or metastatic solid tumors, including ccRCC	Chongqing Precision Biotech Co., Ltd.
NCT05795595	NA	1/2	250	CD-70 CAR-T (CTX131) in relapsed or refractory solid tumors, including ccRCC	CRISPR Therapeutics AG
NCT02830724	NA	1/2	124	CD70 CAR-T in CD70-positive advanced or metastatic solid tumors, including ccRCC	National Cancer Institute

Abbreviations: AXL: AXL receptor tyrosine kinase; CAIX: carbonic anhydrase IX; CAR-T: chimeric antigen receptor T cells; ccRCC: clear cell renal cell carcinoma; CD70: cluster of differentiation 70; CRISPR: clustered regularly interspaced short palindromic repeats; MUC1-C: mucin1 cell surface-associated C-terminal; NA: not available; RCC: renal cell carcinoma; ROR2: receptor tyrosine kinase-like orphan receptor 2; TCR-T: T-cell-receptor-engineered T cells.

**Table 5 cells-13-00034-t005:** Representative early-phase vaccine trials in advanced or metastatic RCC.

NCT Number	Trial Name	Phase	Estimated Patients	Description	Sponsor
NCT00031564	MCC-12207	2	49	B7-1 gene-modified autologous tumor cell vaccine + IL-2 in stage IV RCC	H. Lee Moffitt Cancer Center and Research Institute
NCT00458536	NA	1/2	38	dendritic cell tumor fusions + GM-CSF in stage IV RCC	Beth Israel Deaconess Medical Center
NCT00004880	UCLA-9703025	1	14	dendritic cell vaccine + nephrectomy in advanced RCC	Jonsson Comprehensive Cancer Center
NCT00085436	DMS-0238	2	18	dendritic cell vaccine + IL-2 + IFNα-2a in metastatic RCC	Dartmouth-Hitchcock Medical Center
NCT02950766	NA	1	19	NeoVax (personalized NeoAntigen cancer vaccine) + ipilimumab in resectable stage III or IV ccRCC	Dana-Farber Cancer Institute
NCT01265368	MGN1601-CT1	1/2	19	genetically modified allogeneic tumor cells vaccine + DNA-based immunomodulator in advanced RCC	Mologen AG
NCT05127824	NA	2	42	TBVA-dendritic cell vaccine + cabozantinib in non-metastatic ccRCC	University of Pittsburgh
NCT05641545	IVAC-RCC-001	1	10	personalized neoantigen vaccine + nivolumab + ipilimumab in advanced or metastatic RCC	SLK Kliniken Heilbronn GmbH
NCT05269381	PNeoVCA	1	36	personalized neoantigen peptide-based vaccine + pembrolizumab advanced solid tumors, including RCC	Mayo Clinic
NCT05329532	ModiFY	1/2	144	Modi-1/Modi-1v vaccine as monotherapy or + pembrolizumab in advanced TNBC, SCCHN, HGSOC, or RCC.	Scancell Ltd.
NCT02432963	NA	1	11	p53MVA vaccine + pembrolizumab in solid tumors failed prior therapy	City of Hope Medical Center

Abbreviations: ccRCC: clear cell renal cell carcinoma; GM-CSF: granulocyte macrophage colony-stimulating factor; HGSOC: high-grade serous ovarian carcinoma; IFNα: interferon-α; p53MVA: modified vaccinia virus Ankara vaccine expressing p53; RCC: renal cell carcinoma; SCCHN: human-papillomavirus-negative squamous cell carcinoma of the head and neck; TBVA: tumor blood vessel antigen; TNBC: triple-negative breast cancer.

## Data Availability

Not applicable.

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
