# Peer review of "Emerging Immunotherapy Approaches for Advanced Clear Cell Renal Cell Carcinoma"

_cells, 2023, doi:10.3390/cells13010034_

Round 1

Reviewer 1 Report

Comments and Suggestions for Authors

This is an interesting review on immunotherapeutical approaches to RCC.

Major comments:
1) Please add a paragraph on the use of CIK cells in renal cell carcinoma, including addition in the figure.

Comments on the Quality of English Language

2) Minor corrections concerning the quality of English language is needed.

Author Response

This is an interesting review on immunotherapeutical approaches to RCC.

Major comments:

1) Please add a paragraph on the use of CIK cells in renal cell carcinoma, including addition in the figure.

Response: We are truly thankful for the reviewer's recommendations. In response, we have introduced a new section, Section 6.3 titled "Adoptive Cell Transfer" (page 18), to our manuscript. This section provides a comprehensive discussion of the application of CIK cells in renal cell carcinoma. Corresponding modifications have been applied to Figure 1 to reflect this addition. The content of the newly added section is as follows::” Current research also includes numerous clinical trials investigating adoptive cell transfer (ACT) methodologies in ccRCC. This strategy involves the introduction of ex vivo activated immune cells such as lymphokine-activated killer (LAK) cells, cytokine-induced killer (CIK) cells, or tumor-infiltrating lymphocytes (TILs) [111]. Recent explorations into combining CIK cells with targeted therapies have yielded optimistic outcomes in RCC treatment.

An illustrative case reported by Yang et al. detailed a stage III clear cell RCC patient treated with autologous CIK cells and sorafenib following surgery [112]. Despite initial severe side effects from sorafenib, a reduced dose alongside CIK cell therapy led to stable disease and improved quality of life. In addition, a retrospective study by Mai et al. assessed 34 patients with metastatic RCC, comparing the efficacy of sunitinib or sorafenib alone versus in combination with dendritic cell-CIK (DC-CIK) cells [113]. The combination therapy showed significantly better PFS and three-year OS, with fewer disease progressions and deaths. A phase I trial also examined the safety and efficacy of pembrolizumab, a PD-1 inhibitor, combined with DC-CIK cells in various solid tumors, including RCC [114]. This combination led to a median OS of 270 days and PFS of 162 days, with a 75% disease control rate in RCC patients. Adverse reactions were generally reversible and manageable.

Building upon this momentum, chimeric antigen receptor natural killer (CAR-NK) and chimeric antigen receptor natural killer T (CAR-NKT) cells are being recognized as formidable additions to the ACT spectrum [115-117]. Their inherent capabilities for immediate tumor recognition and destruction, coupled with a lower likelihood of adverse effects, mark them as promising contenders for future ccRCC therapies [118, 119]. The exploration of these cell-based modalities could greatly influence the next generation of treatments for ccRCC, enhancing efficacy while minimizing toxicity.

Such innovative ACT approaches represent a promising avenue in the treatment of ccRCC, offering potential for more effective and targeted immune responses against cancer cells. The integration of these novel therapies could potentially redefine the therapeutic landscape for patients with ccRCC.”

2) Minor corrections concerning the quality of English language is needed.

Response: We appreciate this suggestion and have taken additional steps to enhance the quality of the manuscript's language. This includes rectifying typographical errors, refining punctuation, and revising certain descriptions across the manuscript.

Reviewer 2 Report

Comments and Suggestions for Authors

Lingbin Meng and colleagues present a quality and well-written review manuscript focused on emerging immunotherapy approaches for advanced clear cell renal cell carcinoma.

Authors provide a comprehensive overview of the evolving immunotherapeutic landscape for metastatic ccRCC. Immune checkpoint inhibitors like PD-1/PD-L1 and CTLA-4 inhibitors have demonstrated clinical efficacy as monotherapies and in combination regimens. Combination immunotherapies pairing ICIs with antiangiogenic agents, other immunomodulators, or novel therapeutic platforms such as bispecific antibodies and CAR-T cell therapy, are areas of active research. Beyond checkpoint blockade, additional modalities including therapeutic vaccines, cytokines, and oncolytic viruses are also being explored for ccRCC. 

Authors discuss the mechanisms, major clinical trials, challenges, and future directions for these emerging immunotherapies. While current strategies have shown promise in improving patient outcomes, continued research is critical to expand and optimize immunotherapy approaches for advanced ccRCC. Realizing the full potential of immunotherapy will require elucidating mechanisms of response and resistance, developing predictive biomarkers, and rationally designing combination therapeutic regimens tailored to individual patients. Advances in immunotherapy carry immense promise for transforming the management of metastatic ccRCC.

Authors cover such aspects as: immune checkpoint inhibitors, combination immunotherapies, CAR-T cell therapy, other immune-based therapies (such as vaccines, cytokines, oncolytic viruses). 

Finally, authors conclude that the rise of immunotherapy as a transformative force in ccRCC is undeniable. Realizing the full potential of these promising therapies demands continued collaborative efforts, translating insights from bench to bedside, and careful attention to ethical considerations. As scientists unravel the intricate interplay between ccRCC and the immunsystem, scientists edge closer to a new era of groundbreaking immunotherapies that instill hope in patients battling this challenging disease.

Overall, the manuscript is highly valuable for the scientific community and should be accepted for publication after minor edits are made.

=======================================

Other comments:

1) Please check for typos and punctuation throughout the manuscript.

2) Please improve figures/tables where appropriate.

3) With regards to the use of CAR-T cells against solid tumors - authors are kindly encouraged to cite the following article that reports the the development of solid tumor models for assessment of CAR-T cells activity. DOI: 10.3390/biomedicines11020626

Author Response

1) Please check for typos and punctuation throughout the manuscript.

Response: We are grateful for the reviewer's positive feedback and appreciate this suggestion. We have taken additional steps to enhance the quality of the manuscript's language. This includes rectifying typographical errors, refining punctuation, and revising certain descriptions across the manuscript.

2) Please improve figures/tables where appropriate.

Response: We thank the reviewer for this suggestion. Per request from another reviewer, we have included a new section discussing Adoptive Cell Transfer in RCC, which has been integrated into the Figure 1.

3) With regards to the use of CAR-T cells against solid tumors - authors are kindly encouraged to cite the following article that reports the the development of solid tumor models for assessment of CAR-T cells activity. DOI: 10.3390/biomedicines11020626

Response: We really appreciate this suggestion and have accordingly inserted the citation (reference 90) into the sentence on page 14 that reads, “However, ongoing investigation is evaluating the efficacy of this therapy in solid tumors, such as ccRCC [89], alongside advancements in creating solid tumor models to evaluate CAR T cell activity [90].”

Reviewer 3 Report

Comments and Suggestions for Authors

The authors have presented a comprehensive review of the current state of immunotherapy approaches for advanced renal cell carcinoma. They summarize both immune checkpoint inhibitor monotherapies and combination immunotherapies  with antiangiogenic agents. They cover novel therapies currently being investigated including CAR T cell therapy and bispecific antibodies as well as therapeutic vaccines, cytokines and oncolytic viruses. Each section includes a background paragraph to explain how the particular therapy targets cancer cells, which is helpful to the reader who is less familiar with some of the newer approaches. The authors  include comprehensive tables that summarize completed and on-going clinical trials and include a synopsis of the outcomes for comparison. All pertinent references are included. The review is  organized in a logic manner and very well written.  Overall, this manuscript provides a comprehensive and timely review of immunotherapy approaches in the setting of advanced renal cell carcinoma.

Author Response

Thank you for your encouraging and detailed feedback on our manuscript. Your positive comments motivate us to continue our work in this field, and we are grateful for your thorough review and constructive feedback.

Round 2

Reviewer 1 Report

Comments and Suggestions for Authors

The authors added a third affiliation. Please add which person has this affiliation.

Comments on the Quality of English Language

Minor corrections.